# GhostSR: Learning Ghost Features for Efficient Image Super-Resolution

**Ying Nie**                                                    *ying.nie@huawei.com*
*Huawei Noah's Ark Lab*

**Kai Han**                                                     *kai.han@huawei.com*
*Huawei Noah's Ark Lab*

**Zhenhua Liu**                                                 *liu.zhenhua@huawei.com*
*Huawei Noah's Ark Lab*

**Chuanjian Liu**                                              *liuchuanjian@huawei.com*
*Huawei Noah's Ark Lab*

**Yunhe Wang**                                                 *yunhe.wang@huawei.com*
*Huawei Noah's Ark Lab*

**Reviewed on OpenReview:** *https://openreview.net/forum?id=tbd9f3HwPy*

## Abstract

Modern single image super-resolution (SISR) systems based on convolutional neural networks (CNNs) have achieved impressive performance but require huge computational costs. The problem on feature redundancy has been well studied in visual recognition task, but rarely discussed in SISR. Based on the observation that many features in SISR models are also similar to each other, we propose to use shift operation for generating the redundant features (*i.e.* ghost features). Compared with depth-wise convolution which is time-consuming on GPU-like devices, shift operation can bring a real inference acceleration for CNNs on common hardware. We analyze the benefits of shift operation in SISR and make the shift orientation learnable based on the Gumbel-Softmax trick. Besides, a clustering procedure is explored based on pre-trained models to identify the intrinsic filters for generating corresponding intrinsic features. The ghost features will be generated by moving these intrinsic features along a certain orientation. Finally, the complete output features are constructed by concatenating the intrinsic and ghost features together. Extensive experiments on several benchmark models and datasets demonstrate that both the non-compact and lightweight SISR CNN models embedded with the proposed method can achieve a comparable performance to the baseline models with a large reduction of parameters, FLOPs and GPU inference latency. For example, we reduce the parameters by 46%, FLOPs by 46% and GPU inference latency by 42% of ×2 EDSR model with almost lossless performance. Code will be available at `https://gitee.com/mindspore/models/tree/master/research/cv/GhostSR`.

## 1 Introduction

Single image super-resolution (SISR) is a classical low-level computer vision task, which aims at recovering a high-resolution (HR) image from its corresponding low-resolution (LR) image. Since multiple HR images could be down-sampled to the same LR image, SISR is an ill-posed reverse problem. Recently, deep convolutional neural network (CNN) based methods have made significant improvement on SISR task through carefully designed network architectures. The pioneer work SRCNN (Dong et al., 2014) which contains only three convolutional layers outperforms the previous non-deep learning methods by a large margin. Subsequently, the capacity of CNNs is further excavated with deeper and more complex architectures (Kim et al.,

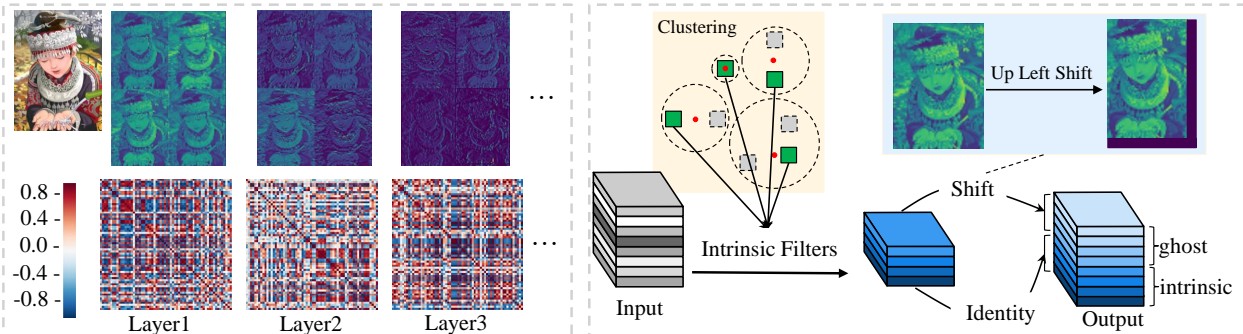

Figure 1: Visualization of features generated by different layers in VDSR (Kim et al., 2016a), which obviously has many similar features (Left). The heatmaps of features' cosine similarity also indicate the phenomenon of redundancy. The redundant features (*i.e.* ghost features) can be generated by cheap operation such as shift based on intrinsic features. The intrinsic features are generated by intrinsic filters, which are selected via clustering pre-trained model (Right). The red dots and green rectangles in clustering represent the cluster centers and the selected intrinsic filters, respectively. Shifting to the upper left is taken as an example to visualize the shift operation.

2016a; Lim et al., 2017; Zhang et al., 2018b;c), which significantly improve the performance. However, these networks usually involve a large number of parameters and floating-point operations (FLOPs), limiting its deployment on portable devices like mobile phones and embedded devices.

Many works have been proposed on model compression for visual classification task, including lightweight architecture design (Howard et al., 2017; Zhang et al., 2018a; Ma et al., 2019; Han et al., 2020), pruning (Han et al., 2016; Li et al., 2017), and quantization (Zhou et al., 2016; Nie et al.; Li et al., 2021), *etc.* Wherein, GhostNet (Han et al., 2020) makes a deep analysis of feature redundancy in the neural network on classification task, and then proposes to generate the redundant features (*i.e.* ghost features) with cheap operations based on the intrinsic features. In practice, the intrinsic features are generated by the regular convolution operation (*i.e.* expensive operation), and then the depth-wise convolution operation (*i.e.* cheap operation) is employed on the intrinsic features for generating the ghost features. Finally, the complete output features are constructed by concatenating the intrinsic and ghost features together. GhostNet achieves competitive accuracy on the ImageNet dataset with fewer parameters and FLOPs. However, there are two major drawbacks in GhostNet that prevent its successful application to other vision tasks, especially the SISR task. Firstly, the so-called cheap operation is not cheap at all for the commonly used GPU-like devices. That is, depth-wise convolution can not bring practical speedup on GPUs due to its low arithmetic intensity (ratio of FLOPs to memory accesses) (Wu et al., 2018; Zhang et al., 2018a). In addition, GhostNet simply divides the output channel evenly into two parts: one part for generating intrinsic features and the other part for generating ghost features, which does not take into account the prior knowledge in the pre-trained model.

Compared with visual classification networks, the SISR networks tend to involve more number of FLOPs. For example, ResNet-50 He et al. (2016) and EDSR Lim et al. (2017) are typical networks in classification task and super-resolution task, respectively. Correspondingly, the FLOPs required for processing a single $3 \times 224 \times 224$ image using ResNet-50 and $\times 2$ EDSR are 4.1G and 2270.9G, respectively. Therefore, the compression and acceleration of SISR networks is more urgent and practical. To date, many impressive works have been proposed for compressing and accelerating the SISR networks (Ahn et al., 2018; Song et al., 2020; Hui et al., 2018; Zhao et al., 2020; Zhang et al., 2021b;a; Li et al., 2022). However, the basic operations in these works are still conventional convolution operation, which do not consider the redundancy in features and the practical speed in common hardware. Since most of the existing SISR models need to preserve the overall texture and color information of the input images, there are inevitably many similar features in each layer as observed in the left of Figure 1. In addition, considering that the size of images displayed on modern intelligent terminals are mostly in high-resolution format (2K-4K), the efficiency of SISR models should be maintained on the common platforms with large computing power (*e.g.* , GPUs and NPUs).

Considering that the phenomenon of feature redundancy also exists in the SISR models, in this paper, we introduce a GhostSR method to generate the redundant features with a real cheap operation, *i.e.* shift operation (Wu et al., 2018; Jeon & Kim, 2018; Chen et al., 2019). Compared with the depth-wise convolution in GhostNet (Han et al., 2020), shift operation can bring practical speedup in common hardware with negligible FLOPs and parameters. Specifically, the shift operation moves the intrinsic features along a specific orientation for generating the ghost features. We conclude two benefits of the shift operation for the SISR task, including the ability to enhance high frequency information and the ability to enlarge the receptive field of filters. In addition, the Gumbel-Softmax trick (Jang et al., 2016) is exploited so that each layer in the neural network can adaptively learns its own optimal moving orientation of the intrinsic features for generating the ghost features. The complete output features are then constructed by concatenating the intrinsic and ghost features together. With the efficient CUDA implementation of shift operation, GhostSR brings a practical acceleration on GPUs during inference. Last but not least, the prior knowledge in the pre-trained model is taken into account in distinguishing which part is intrinsic features and which part is ghost features. Specifically, the intrinsic filters are identified via clustering the weights of a pre-trained model based on output channel, and the features generated by the intrinsic filters are taken as intrinsic features. The other part is taken as the ghost features. The extensive experiments conducted on several benchmarks demonstrate the effectiveness of the proposed method.

## 2 Related Works

### 2.1 Model Compression

In order to compress deep CNNs, a series of methods have been proposed which include lightweight architecture design (Howard et al., 2017; Zhang et al., 2018a; Ma et al., 2019; Han et al., 2020), network pruning (Han et al., 2016; Li et al., 2017; Ding et al., 2019), knowledge distillation (Hinton et al., 2015; Romero et al., 2015) and low-bit quantization (Courbariaux et al., 2015; Zhou et al., 2016; Nie et al.; Li et al., 2021), *etc.* Han *et al.* (Han et al., 2016) remove the connections whose weights are lower than a certain threshold in network. Ding *et al.* (Ding et al., 2019) propose a centripetal SGD optimization method for network slimming. Hinton *et al.* (Hinton et al., 2015) introduce the knowledge distillation scheme to improve the performance of student model by inheriting the knowledge from a teacher model. Courbariaux *et al.* (Courbariaux et al., 2015) quantize the weights and activations into 1-bit value to maximize the compression and acceleration of the network. In addition, lightweight network architecture design has demonstrated its advantages in constructing efficient neural network architectures. MobileNets (Howard et al., 2017; Sandler et al., 2018) are a series of lightweight networks based on bottleneck structure. Liu *et al.* (Liu et al., 2022) propose to enlarge the capacity of CNN models by fine-grained FLOPs allocation for the width, depth and resolution on the stage level. Wu *et al.* (Wu et al., 2018) first propose the shift operation which moves the input features horizontally or vertically, then Jeon *et al.* (Jeon & Kim, 2018) and chen *et al.* (Chen et al., 2019) further make the shift operation learnable in visual classification task. ShiftAddNet You et al. (2020) and ShiftAddNAS You et al. (2022) leverage the bit-wise shift for building energy-efficient networks. Recently, Han *et al.* (Han et al., 2020) analyze the redundancy in features and introduce a novel GhostNet module for building efficient models. In practice, GhostNet generates part of features with depth-wise convolution, achieving satisfactory performance in high-level tasks with fewer parameters and FLOPs. However, due to the fragmented memory footprints raised by depth-wise convolution, GhostNet can not bring a practical acceleration on common GPU-like devices.

### 2.2 Efficient Image Super-Resolution

Numerous milestone works based on convolutional neural network have been proposed on image super-resolution task (Dong et al., 2014; Kim et al., 2016a; Tai et al., 2017b; Lim et al., 2017; Zhang et al., 2018c;b). However, these works are difficult to deploy on resource-limited devices due to their heavy computation cost and memory footprint. To this end, model compression on super-resolution task is attracting widespread attention. FSRCNN (Dong et al., 2016) first accelerate the SISR network by a compact hourglass-shape architecture. DRCN (Kim et al., 2016b)and DRRN (Tai et al., 2017a) adopt recursive layers to build deep network with fewer parameters. CARN (Ahn et al., 2018) reduce the computation overhead by combining the

efficient residual block with group convolution. IMDN (Hui et al., 2019) construct the cascaded information multi-distillation block for efficient feature extraction. LatticeNet (Luo et al., 2020) utilize series connection of lattice blocks and the backward feature fusion for building a lightweight SISR model. SMSR (Wang et al., 2021) develop a sparse mask network to learn sparse masks for pruning redundant computation. Attention mechanism is also introduced to find the most informative region to reconstruct high-resolution image with better quality (Zhao et al., 2020; Muqeet et al., 2020; Zhang et al., 2018b; Magid et al., 2021; Niu et al., 2020). To improve the performance of lightweight networks, distillation has been excavated to transfer the knowledge from experienced teacher networks to student networks (Hui et al., 2018; Gao et al., 2018; Lee et al., 2020b). Neural architecture search (NAS) is also employed to exploit the efficient architecture for image super-resolution task (Song et al., 2020; Guo et al., 2020; Lee et al., 2020a; Zhan et al., 2021). Recently, based on the motivation that different image regions have different restoration difficulties and can be processed by networks with different capacities, ClassSR (Kong et al., 2021) and FAD (Xie et al., 2021) propose dynamic super-resolution networks. However, the dynamic SISR networks equipped with multiple processing branches usually contain several times the number of parameters of the original network, which limits its deployment on portable devices.

## 3 Approach

In this section, we describe the details of the proposed GhostSR method for efficient image super-resolution.

### 3.1 Shift for Generating Ghost Features

The CNN-based super-resolution models consist of massive convolution computations. For a vanilla convolutional layer, producing the output features $Y \in \mathbb{R}^{c_o \times h \times w}$ requires $hwc_oc_is^2$ FLOPs where $c_o$, $h$, $w$, $c_i$, $s \times s$ are the number of output channels, the height of output, the width of output, the number of input channels and the kernel size. The computational cost of convolution consumes much energy and inference time. On the other hand, we observe that some features in SISR network is similar to another ones, that is, these features can be viewed as redundant versions of the other intrinsic features, as shown in the left of Figure 1. We term these redundant features as ghost features. In fact, ghost features can provide a wealth of texture and high-frequency information, which cannot be directly removed (Han et al., 2020; Yuan et al., 2020). Instead of discarding, we propose to utilize a more efficient operator, *i.e.* , shift operation, to generate them.

Assuming that the ratio of ghost features is $\lambda$ where $0 \leq \lambda < 1$, then the number of intrinsic and ghost features is $(1 - \lambda)c_o$ and $\lambda c_o$, respectively. The intrinsic features $I \in \mathbb{R}^{(1-\lambda)c_o \times h \times w}$ are generated by regular convolution, and the ghost features $G \in \mathbb{R}^{\lambda c_o \times h \times w}$ are generated by shift operation based on $I$ since shift is cheap yet has many advantages on super-resolution task, which will be discussed in detail later. Formally, the vertical and horizontal offsets to be shifted are $i_o$ and $j_o$, where $-d \leq i_o \leq d$, $-d \leq j_o \leq d$, and $d$ is the maximum offset, then the element in position $(y, x)$ of $R$ can be obtained as:

$$G_{y,x,c_1} = \sum_{i=-d}^{d} \sum_{j=-d}^{d} I_{y+i,x+j,c_2} W_{i,j}, \tag{1}$$

where $W \in \{0, 1\}^{(2d+1) \times (2d+1)}$ is a one-hot matrix denoting the offset values, and $c_1$ and $c_2$ are the channel index of ghost features and the corresponding index of intrinsic features, respectively. All the elements in $W$ are 0 except that the value in the offset position is 1:

$$W_{i,j} = \begin{cases} 1, & \text{if } i = i_o \text{ and } j = j_o, \\ 0, & \text{otherwise.} \end{cases} \tag{2}$$

Finally, we concatenate the intrinsic and ghost features together as the complete output: $O = [I, G]$. Compared to original convolution layer, our method cuts the FLOPs directly by a ratio of $\lambda$ since shift operation is FLOPs-free. More importantly, with the efficient CUDA implementation of shift operation, we can bring a practical inference acceleration on GPU-like devices.

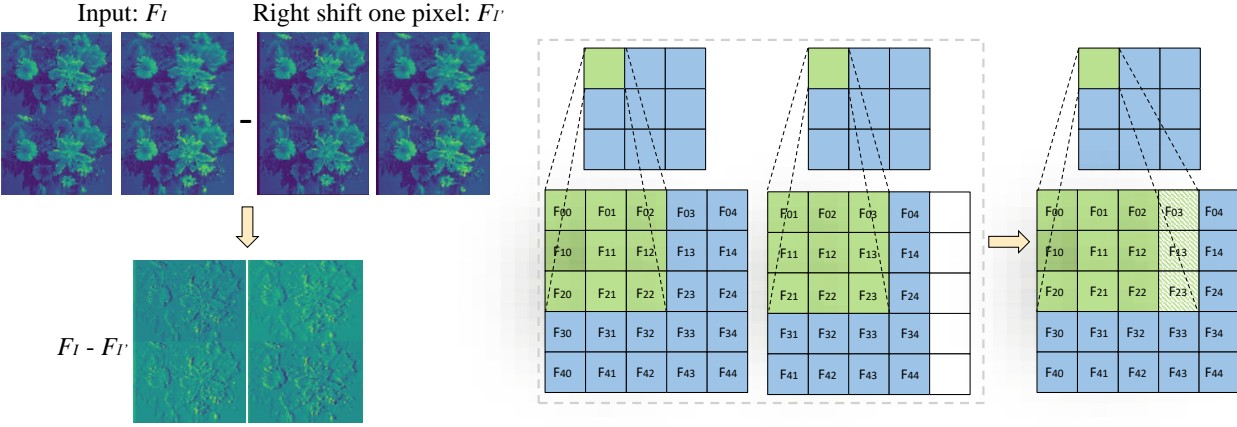

(a) Enhance high frequency information            (b) Enlarge the receptive field of filters

Figure 2: Benefits of shifting the features for SISR task.

### 3.1.1  Benefits of Shift for Super-resolution.

Super-resolution aims at recovering a high-resolution image from its corresponding low-resolution image. The enhanced high frequency information such as texture could be helpful for improving the quality of recovered high-resolution image (Zhang et al., 2018b; Zhou et al., 2018). Given a input feature map $F_I$, we shift $F_I$ one pixel to the right across all channels, and pad the vacant position with zeros to get the shifted feature maps $F_{I'}$. The texture information can be enhanced by $F_I - F_{I'}$, as shown in the left of Figure 2. In convolution neural network, the vanilla features and the shifted features are concatenated together to be processed by next layer's convolution operation. The convolution operation can be seen as a more complex operation involving subtraction, which can enhance the high frequency information to some extent.

In addition, the combination of two spatially dislocated feature maps can enlarge the receptive field of CNNs, which is critical for super-resolution task (Wang et al., 2019; He et al., 2019b). In other words, the shift operation of feature maps provides a spatial information communication of convolution filters. For instance, as demonstrated in the right of Figure 2, when shift a feature map one pixel to the left, the receptive field of the same location on the next layer's feature map shift one pixel to the left correspondingly. The convolution operation performed on the combination of dislocation feature maps results in a wider receptive field.

Last but not least, as opposed to the low arithmetic intensity (ratio of FLOPs to memory accesses) of depth-wise convolution, shift operation is more efficient in terms of practical speed, which will be compared in detail in the experiments section.

### 3.2  Make the Shift Learnable

During the training process, the shift operation in Eq. 1 can be implemented by a special case of depth-wise convolution where only one weight is 1 and the others are 0. Figure 3 gives an example to illustrate how the shift operation works. In order to flexibly adjust the offset of intrinsic features during training, the offset weight $W$ need to be learnable. However, the one-hot values in $W$ make it difficult to optimize the weights. Therefore, the Gumbel-Softmax trick (Jang et al., 2016) is adopted for addressing this issue. The Gumbel-Softmax trick feed-forwards the one-hot signal and back-propagates the soft signal, which solves the non-derivableness of sampling from categorical distribution.

We create a proxy soft weight $W' \in \mathbb{R}^{(2d+1) \times (2d+1)}$ for representing the inherent values of one-hot $W$. A noise $N \in \mathbb{R}^{(2d+1) \times (2d+1)}$ is randomly sampled from the Gumbel distribution:

$$N_{i,j} = -\log(-\log(U_{i,j})), \tag{3}$$

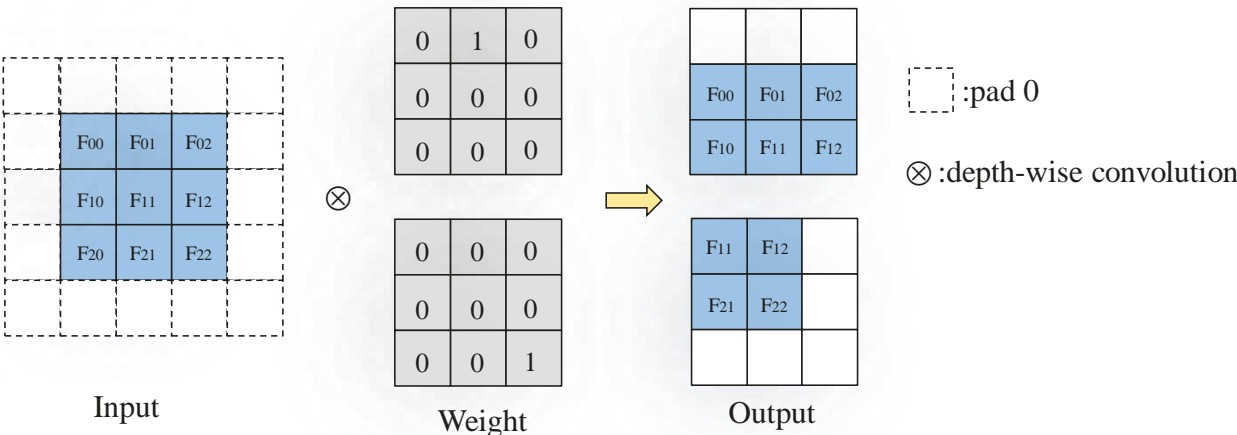

Figure 3: An example of the shift operation during training, which is implemented by a special case of depth-wise convolution where only one weight is 1 and the others are 0.

where $U_{i,j} \sim U(0,1)$ is sampled from uniform distribution. The one-hot weight $W$ is relaxed as

$$S(W'_{i,j}) = \frac{e^{(W'_{i,j}+N_{i,j})/\tau}}{\sum_{i=-d}^{d}\sum_{j=-d}^{d}e^{(W'_{i,j}+N_{i,j})/\tau}}, \tag{4}$$

where $\tau$ is the temperature to control the sharpness of the softmax function, and the function approximates the discrete categorical sampling. Then, we can obtain the values of offset indices as

$$i_o, j_o = \arg\max_{i,j} S(W'). \tag{5}$$

During feed-forward process, the values of $W$ can be computed as Eq. 2. As for the back-propagation process, a straight-through estimator is utilized, that is, the derivative $\frac{\partial W}{\partial W'}$ is approximated calculated using the derivative of Eq. 4:

$$\frac{\partial W}{\partial W'} = \frac{\partial S(W')}{\partial W'}. \tag{6}$$

Then, the trainable shift weight $W$ and other trainable parameters can be trained end-to-end. After training, we pick the position of the maximum value as the shift offset position and construct the inference graph of GhostSR networks.

### 3.3 Intrinsic Features in Pre-trained Model

In the proposed method, we first generate $(1-\lambda)c_o$ features as intrinsic features, then use the shift operation to generate the other features as ghost features based on the intrinsic features, and finally concatenate the intrinsic and ghost features together as the complete output features. If we train a GhostSR model from scratch, the indices $c_1$ and $c_2$ in Eq. 1 are set simply by order. If a pre-trained vanilla SR model is provided, we can utilize the relation of intrinsic and ghost features for better performance. Since the goal of this work is to reduce the inference latency, parameters and FLOPs of SISR models, and some other works such as pruning or quantization (Li et al., 2021; Han et al., 2016) also require pre-trained models, pre-training here is reasonable.

Given a pre-trained SR model, we aim to replace part of the convolution operations with shift operations for generating ghost features. However, for the output features of a certain layer in network, it is not clear which part is intrinsic and which part is ghost. We address this problem by clustering the filters in pre-trained models, and the features generated by the filter which is nearest to the cluster centroid are taken as intrinsic features. Specifically, the weights of all convolution filters are firstly vectorized from $[c_o, c_i, s, s]$ to $[c_o, c_i \times s \times s]$ and obtain weight vectors $\{f_1, f_2, \cdots, f_{c_o}\}$. Then the weight vectors are divided into $(1-\lambda)c_o$

clusters $\mathcal{G} = \{G_1, G_2, ..., G_{(1-\lambda)c_o}\}$. Any pair of points in one cluster should be as close to each other as possible:

$$\min_{\mathcal{G}} \sum_{k=1}^{(1-\lambda)c_o} \sum_{i \in G_k} ||f_i - \mu_k||_2^2, \tag{7}$$

where $\mu_k$ is the mean of points in cluster $G_k$. We use clustering method $k$-means for clustering filters with the objective function in Eq. 7. For the clusters which contain only one filter, we take this filter as intrinsic filter. For the clusters which contain multiple filters, the centroid may not really exist in the original weight kernels, so we select the filter which is nearest to the centroid as intrinsic filter, and the features generated by the intrinsic filters are taken as intrinsic features. The index of intrinsic filters can be formulated as

$$\mathbb{I}_k = \begin{cases} i \in G_k, & \text{if } |G_k| = 1, \\ \underset{i \in G_k}{\arg\min} \, ||f_i - \mu_k||_2^2, & \text{otherwise.} \end{cases} \tag{8}$$

The set of intrinsic indices is $\mathbb{I} = \{\mathbb{I}_1, \mathbb{I}_2, \cdots, \mathbb{I}_{(1-\lambda)c_o}\}$ whose corresponding filters are preserved as convolution operations, and the other filters are replaced by the proposed shift operations.

After finding the intrinsic filters in each layers of the pre-trained model, we assign the corresponding weight in pre-trained model to the intrinsic filters. Thus, we can maximally utilize the information in pre-trained model to identify intrinsic features and inherit pre-trained filters for better performance.

## 4 Experiments

In this section, we conduct extensive experiments on non-compact and lightweight networks. The detailed quantitative and qualitative evaluations are provided to verify the effectiveness of the proposed method.

### 4.1 Experimental Settings

#### 4.1.1 Datasets and Metrics.

To evaluate the performance of the method, following the setting of (Lim et al., 2017; Zhang et al., 2018c; Ahn et al., 2018), we use 800 images from DIV2K (Timofte et al., 2017) dataset to train our models. In order to compare with other state-of-the-art methods, we report our result on four standard benchmark datasets: Set5 (Bevilacqua et al., 2012), Set14 (Zeyde et al., 2010), B100 (Martin et al., 2001) and Urban100 (Huang et al., 2015). The LR images are generated by bicubic down-sampling and the SR results are evaluated by PSNR and SSIM (Wang et al., 2004) on Y channel of YCbCr space.

#### 4.1.2 Training Details.

We use four famous SISR models as our baselines: EDSR (Lim et al., 2017), RDN (Zhang et al., 2018c), CARN (Ahn et al., 2018) and IMDN (Hui et al., 2019). These models have various numbers of parameters ranging from 0.69M to 43.71M (million). To maintain the performance of the models embedded in the proposed method, we do not replace the regular convolution in the first and the last layers in these networks (Courbariaux et al., 2015; Zhou et al., 2016), and the point-wise convolution is also kept unchanged if encountered in the middle layers. The detailed architectures are summarized in the Appendix. In addition, unless otherwise specified, the ratio $\lambda$ of ghost features is set to 0.5, the temperature $\tau$ in Eq. 4 is set to 1, and the maximum offset $d$ in Eq. 1 is set to 1.

During training, we crop 16 images with $48 \times 48$ patch size from the LR images on every card for training. The input examples are augmented by random horizontal flipping and 90° rotating. In addition, all the images are pre-processed by subtracting the mean RGB value of the DIV2K dataset. To optimize the model, we use ADAM optimizer (Kingma & Ba, 2014) with $\beta_1 = 0.9$, $\beta_2 = 0.999$, and $\epsilon = 10^{-8}$. We train EDSR and RDN for 300 epochs by single-scale training scheme, and train CARN and IMDN for 1200 epochs by multi-scale and single-scale training scheme respectively. The initial learning rate is set to 1e-4 for all models and reduced by cosine learning rate decay (Zhao et al., 2020; Kong et al., 2021).

Table 1: Quantitative results of baseline convolutional networks and their Ghost versions for scaling factor ×2, ×3 and ×4. We assume the HR image size to be 720p (1280×720) to calculate the FLOPs.

| Scale | Model | Type | Params (M) | FLOPs (G) | Set5 PSNR/SSIM | Set14 PSNR/SSIM | B100 PSNR/SSIM | Urban100 PSNR/SSIM |
|---|---|---|---|---|---|---|---|---|
| ×2 | EDSR | Raw | 40.73 | 9389 | 38.22/0.9612 | 33.86/0.9201 | 32.34/0.9018 | 32.92/0.9356 |
| | | Ghost | 21.85 | 5038 | ↓0.01/↓0.0000 | ↑0.07/↑0.0003 | ↓0.00/↓0.0000 | ↓0.04/↓0.0004 |
| | RDN | Raw | 19.27 | 4442 | 38.25/0.9614 | 33.97/0.9205 | 32.34/0.9017 | 32.90/0.9355 |
| | | Ghost | 9.83 | 2265 | ↓0.06/↓0.0002 | ↓0.01/↑0.0005 | ↓0.00/↓0.0000 | ↓0.10/↓0.0009 |
| | CARN | Raw | 1.59 | 223 | 37.88/0.9601 | 33.54/0.9173 | 32.14/0.8990 | 31.96/0.9267 |
| | | Ghost | 1.19 | 130 | ↓0.00/↓0.0000 | ↓0.00/↑0.0002 | ↓0.03/↓0.0003 | ↓0.09/↓0.0009 |
| | IMDN | Raw | 0.69 | 160 | 37.91/0.9595 | 33.59/0.9170 | 32.15/0.8988 | 32.14/0.9275 |
| | | Ghost | 0.40 | 91 | ↓0.05/↓0.0002 | ↓0.18/↓0.0008 | ↓0.05/↓0.0006 | ↓0.23/↓0.0021 |
| ×3 | EDSR | Raw | 43.68 | 4471 | 34.67/0.9293 | 30.56/0.8466 | 29.26/0.8094 | 28.78/0.8649 |
| | | Ghost | 24.80 | 2541 | ↓0.03/↓0.0004 | ↓0.06/↓0.0007 | ↓0.03/↓0.0005 | ↓0.09/↓0.0018 |
| | RDN | Raw | 19.37 | 1981 | 34.69/0.9296 | 30.54/0.8467 | 29.26/0.8093 | 28.79/0.8654 |
| | | Ghost | 9.92 | 1016 | ↓0.09/↓0.0009 | ↓0.04/↓0.0004 | ↓0.04/↓0.0008 | ↓0.15/↓0.0030 |
| | CARN | Raw | 1.59 | 119 | 34.35/0.9266 | 30.32/0.8415 | 29.08/0.8043 | 28.05/0.8499 |
| | | Ghost | 1.19 | 77 | ↓0.08/↓0.0004 | ↓0.02/↓0.0005 | ↓0.01/↓0.0004 | ↓0.08/↓0.0019 |
| | IMDN | Raw | 0.70 | 72 | 34.32/0.9260 | 30.31/0.8410 | 29.07/0.8037 | 28.15/0.8511 |
| | | Ghost | 0.41 | 41 | ↓0.14/↓0.0012 | ↓0.13/↓0.0024 | ↓0.07/↓0.0013 | ↓0.22/↓0.0044 |
| ×4 | EDSR | Raw | 43.09 | 2896 | 32.46/0.8984 | 28.81/0.7874 | 27.73/0.7416 | 26.60/0.8019 |
| | | Ghost | 24.21 | 1808 | ↓0.04/↓0.0005 | ↓0.05/↓0.0010 | ↓0.06/↓0.0011 | ↓0.12/↓0.0027 |
| | RDN | Raw | 19.42 | 1146 | 32.49/0.8987 | 28.83/0.7875 | 27.72/0.7416 | 26.59/0.8023 |
| | | Ghost | 9.97 | 602 | ↓0.10/↓0.0011 | ↓0.10/↓0.0024 | ↓0.05/↓0.0020 | ↓0.21/↓0.0063 |
| | CARN | Raw | 1.59 | 91 | 32.16/0.8943 | 28.59/0.7810 | 27.58/0.7355 | 26.03/0.7830 |
| | | Ghost | 1.19 | 68 | ↓0.05/↓0.0007 | ↓0.02/↓0.0008 | ↓0.02/↓0.0007 | ↓0.05/↓0.0020 |
| | IMDN | Raw | 0.72 | 41 | 32.19 / 0.8938 | 28.57/0.7805 | 27.54/0.7344 | 26.03/0.7831 |
| | | Ghost | 0.42 | 24 | ↓0.13/↓0.0019 | ↓0.08/↓0.0021 | ↓0.06/↓0.0017 | ↓0.14/↓0.0044 |

Table 2: Average inference latency (ms) for Urban100 dataset with ×2 scale on a single V100 GPU platform.

| Model | EDSR(Original / GhostSR) | RDN(Original / GhostSR) |
|---|---|---|
| Latency | 720.20 / 420.71 | 450.73 / 304.75 |
| Model | CARN(Original / GhostSR) | IMDN(Original / GhostSR) |
| Latency | 49.52 / 37.26 | 35.05 / 25.64 |

## 4.2 Comparison with Baselines

### 4.2.1 Quantitative Evaluation.

In Table 1, we report the quantitative results of the baseline convolutional networks and the proposed GhostSR versions for scaling factor ×2, ×3 and ×4. The results of raw type are obtained by our re-training, which are close to the results in the original papers. The detailed results of our re-training and the results in the original papers are reported in the Appendix. In addition, following the experimental settings in IMDN (Hui et al., 2019), we report the practical average inference latency in Table 2. From the results in Table 1 and Table 2, we can see that both the non-compact (EDSR and RDN) and lightweight (CARN and

IMDN) SISR models embedded in the GhostSR module can achieve comparable performance to that of their baselines with a large reduction of parameters, FLOPs and GPU inference latency. For instance, we reduce the FLOPs of ×2 EDSR, ×2 RDN, ×2 CARN and ×2 IMDN by 46%, 49%, 42% and 43%, respectively, with little performance loss. Similarly, for different networks, the number of parameters and inference latency are also greatly reduced. We attribute it to the superiority of learnable shift and the clustering excavation of prior information in pre-trained models. Urban100 dataset contains 100 images of urban scenes, which is more challenging than other datasets. Therefore, a slightly higher loss of PSNR is acceptable here.

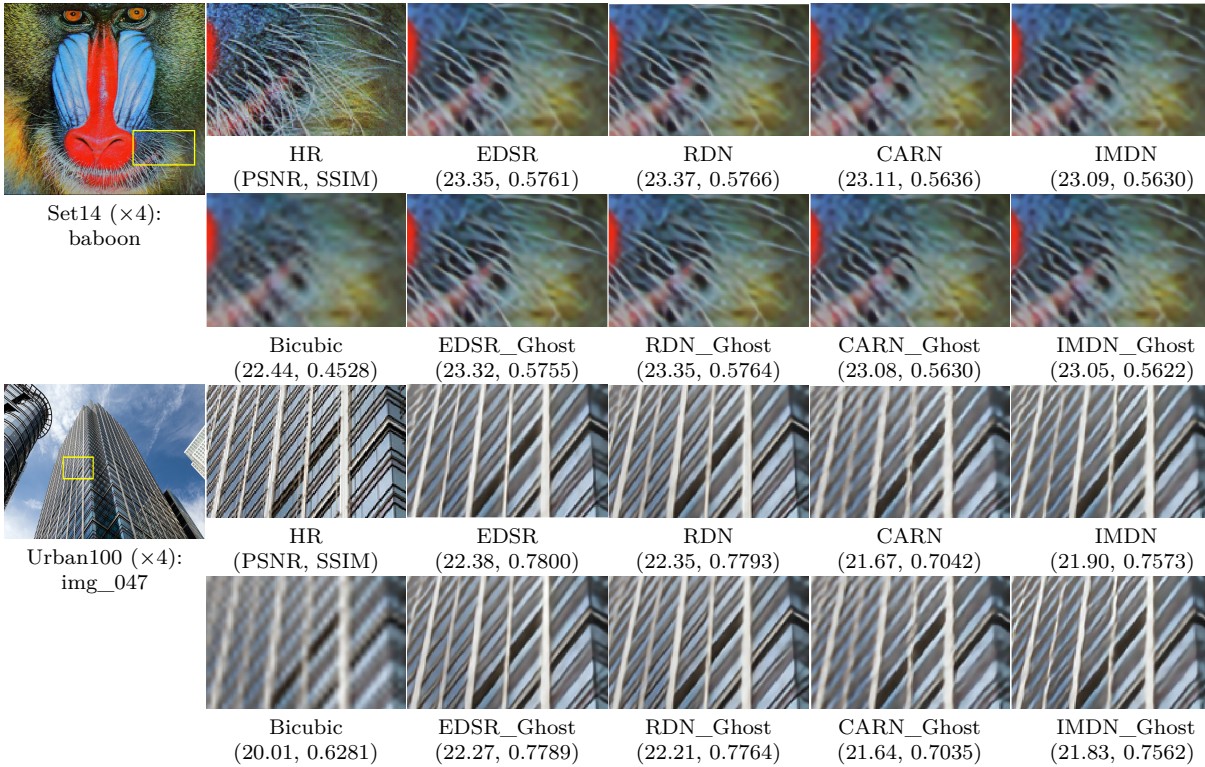

Figure 4: Visual comparisons for ×4 images on Set14 and Urban100 datasets. For the shown examples, the details and textures generated by shift operation are approximately the same as those by regular convolution.

### 4.2.2 Qualitative Evaluation.

The qualitative evaluations on various datasets are shown in Figure 4. We choose the most challenging ×4 task to reveal the difference of visual quality between the original network and the GhostSR versions. From Figure 4, we can see that for both the non-compact and lightweight networks, the details and textures generated by GhostSR are basically the same as those by original network.

### 4.3 Comparison with State-of-the-arts

We compare the proposed GhostSR models with the state-of-the-arts including manual-designed efficient SISR methods (VDSR (Kim et al., 2016a), CARN (Ahn et al., 2018), CARN_M (Ahn et al., 2018), PAN (Zhao et al., 2020), SelNet (Choi & Kim, 2017), OISR (He et al., 2019a), BTSRN (Fan et al., 2017), LapSRN (Lai et al., 2017)) and NAS-based (neural architecture search) efficient SISR methods (ESRN (Song et al., 2020), FALSR (Chu et al., 2019), MoreMNAS (Chu et al., 2020)). From the results in Figure 5, we can see that the GhostSR models achieves a comparable PSNR with less FLOPs and inference latency. Since the attention operation in PAN (Zhao et al., 2020) is very time-consuming, our GhostSR-IMDN and GhostSR-CARN is faster. In addition, PAN achieves a higher PSNR with the use of additional Flickr2K (Lim et al., 2017) training data, which is not used in our method.

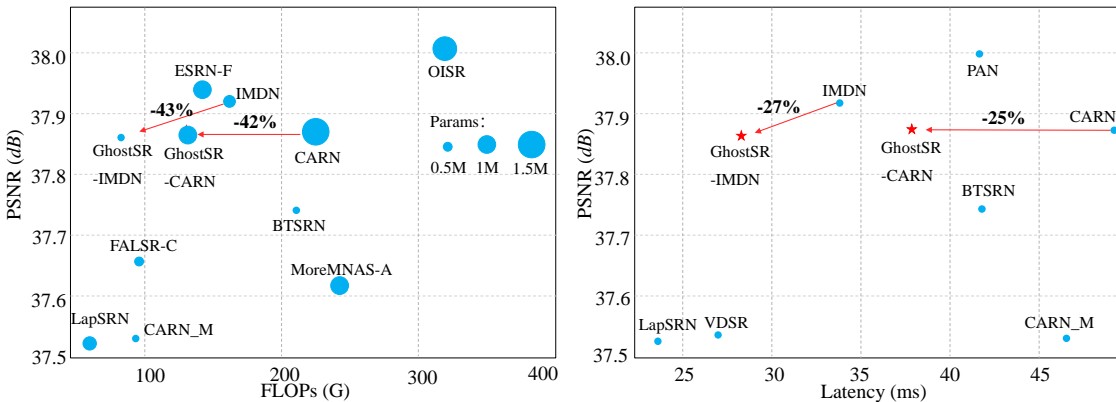

Figure 5: PSNR *v.s.* FLOPs and PSNR *v.s.* Latency. The PSNR is calculated on Set5 dataset with ×2 scale, and the FLOPs is calculated with a 640 × 360 image. The average inference latency is calculated on a single V100 GPU using Urban100 dataset with ×2 scale.

Table 3: Comparison of different ×3 CARN network under similar FLOPs budget. The FLOPs is calculated with a 426×240 image, and the average inference latency is calculated on a single V100 GPU using Urban100 dataset with ×3 scale. The best results are in bold.

| Type | Params | FLOPs | Latency | Set5 | Set14 | B100 | Urban100 |
|------|--------|-------|---------|------|-------|------|----------|
|      | (M)    | (G)   | (ms)    | PSNR/SSIM | PSNR/SSIM | PSNR/SSIM | PSNR/SSIM |
| Original | 1.59 | 119 | 32.09 | 34.35/0.9266 | 30.32/0.8415 | 29.08/0.8043 | 28.05/0.8499 |
| Conv-0.80× | **1.15** | 79 | 29.33 | 34.11/0.9242 | 30.13/0.8390 | 28.89/0.8022 | 27.77/0.8460 |
| C-SGD | **1.15** | 79 | 29.05 | 34.15/0.9248 | 30.16/0.8395 | 28.92/0.8026 | 27.84/0.8467 |
| HRank | 1.19 | 78 | 28.56 | 34.16/0.9251 | 30.14/0.8395 | 28.95/0.8028 | 27.86/0.8471 |
| GhostSR | 1.19 | **77** | **23.28** | **34.27/0.9262** | **30.30/0.8410** | **29.07/0.8039** | **27.97/0.8480** |

## 4.4 Comparison with Other Compression Methods

### 4.4.1 Comparison with Pruning Methods.

Pruning is a classical type of model compression methods which aims to cut out the unimportant channels. Unlike image recognition that requires the high-level coarse feature, the detailed difference between feature maps is important for super-resolution task. Our GhostSR does not remove any channel and re-generate the ghost features by the learnable shift. Thus, we compare GhostSR with the representative network pruning methods including C-SGD (Ding et al., 2019) and HRank (Lin et al., 2020). In table 3, we report the quantitative results of different methods of reducing FLOPs for ×3 CARN network. Conv-0.80× means to directly reduce the number of channels in each layer of the network to 0.80 times the original number. We re-implement the C-SGD (Ding et al., 2019) and HRank (Lin et al., 2020) and apply them for the SISR task. CARN itself is a lightweight network, and directly reducing the width of each layer of the network will bring a greater performance drop. For instance, compared with the original CARN, the PSNR of Conv-0.80× is reduced by $0.28dB$ for ×3 scale on Urban100 dataset. When the pruning methods (C-SGD and HRank) are adopted, the PSNR/SSIM is improved significantly. However, there are still some gaps with the GhostSR method. For example, GhostSR exceeds HRank by $0.11dB$ on Urban100 dataset with a faster inference speed.

Figure 6 visualizes the ×3 images generated by different CARN. The visual quality of the images generated by reducing the number of channels is obviously degraded, and the GhostSR maintains the visual quality of images. We attribute it to the characteristics of shift operation, which can compensate for the performance

degradation caused by the reduction in channels. For example, the left side of the enlarged area of img_008 generated by Conv-0.80× or C-SGD Ding et al. (2019) is blurred compared to that of GhostSR method.

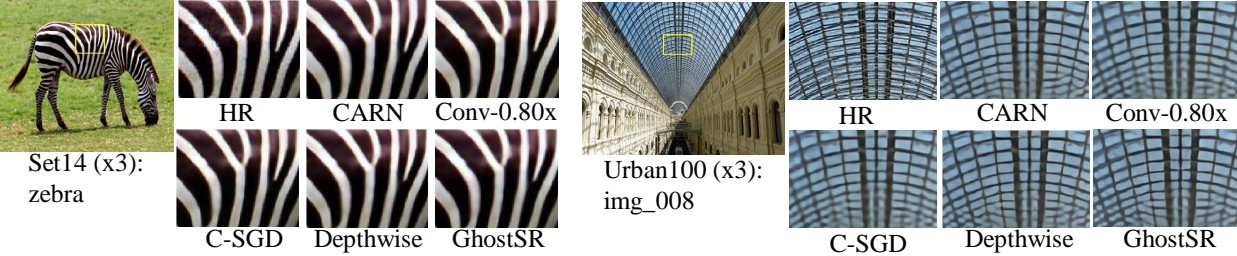

Figure 6: Visual comparisons of different model compression methods for ×3 CARN.

### 4.4.2 Comparison with Depthwise Convolution.

In addition, we also compare the results of replacing the shift operation in our method with the depth-wise convolution, which is denoted by Depthwise in Table 4. The depth-wise convolution is utilized in GhostNet to build efficient models in classification task. When the shift operation in GhostSR model is replaced with depth-wise convolution, there is a slight performance boost due to the introduction of a few more FLOPs. The visual quality between shift and depth-wise is similar in Figure 6. However, the average inference latency of depth-wise operation is greatly increased since it is time-consuming on GPU-like devices due to its fragmented memory footprints. In particular, the latency of Depthwise model is 31.56ms which is 8.28ms more than GhostSR method.

Table 4: Comparison of shift and depth-wise operations on ×3 CARN network.

| Method | FLOPs (G) | Latency (ms) | Set5 PSNR/SSIM | Set14 PSNR/SSIM | B100 PSNR/SSIM | Urban100 PSNR/SSIM |
|---|---|---|---|---|---|---|
| Original | 119 | 32.09 | 34.35 / 0.9266 | 30.32 / 0.8415 | 29.08/0.8043 | 28.05/0.8499 |
| Depthwise | 80 | 31.56 | 34.27 / 0.9263 | 30.31 / 0.8410 | 29.09/0.8042 | 28.00/0.8485 |
| GhostSR | 77 | 23.28 | 34.27 / 0.9262 | 30.30 / 0.8410 | 29.07/0.8039 | 27.97/0.8480 |

### 4.5 Ablation Study

### 4.5.1 Analysis on the Sub-Modules in GhostSR.

We first conduct a series of comparison experiments to analyze the effects of sub-modules in GhostSR, and the results are reported in Table 5. For the option that does not use the learnable shift scheme, we replace the shift operation with a simple copy operation, that is, the ghost features are obtained directly by copying the intrinsic features. For the option that does not use the clustered information in pre-trained models, we train the models from scratch. When neither learnable shift scheme nor the clustering based on pre-trained models is used, the PSNR of GhostSR drops by $0.24dB$ on Urban100 dataset. When only the clustering based on pre-trained models or the learnable shift scheme is used, the PSNR drops by $0.13dB$ and $0.08dB$ on Urban100 dataset, respectively.

Figure 7 visualizes the features generated at the same layer for three versions of CARN: copy operation, learnable shift operation and regular convolution operation. The copy version and the learnable shift version of CARN are both trained based on the pre-trained model using the aforementioned clustering procedure. The features of img_001 and lenna are generated in the first and third residual-block of CARN, respectively. From Figure 7, the learnable shift operation extracts textures similar to that of the regular convolution, and more than the copy operation. For example, the edge and texture of features in lenna generated by shift are clearer and richer than those by copy.

Table 5: Analysis on the sub-modules in GhostSR for ×3 CARN network.

| Architecture | | Set5 | Set14 | B100 | Urban100 |
|---|---|---|---|---|---|
| Learnable Shift (Eq. 4, 5) | Clustering (Eq. 7, 8) | PSNR/SSIM | PSNR/SSIM | PSNR/SSIM | PSNR/SSIM |
| × | × | 34.04/0.9241 | 30.06/0.8378 | 28.80/0.8012 | 27.73/0.8439 |
| × | ✓ | 34.17/0.9250 | 30.23/0.8399 | 28.96/0.8023 | 27.84/0.8463 |
| ✓ | × | 34.20/0.9253 | 30.21/0.8397 | 29.00/0.8031 | 27.89/0.8471 |
| ✓ | ✓ | 34.27/0.9262 | 30.30/0.8410 | 29.07/0.8039 | 27.97/0.8480 |

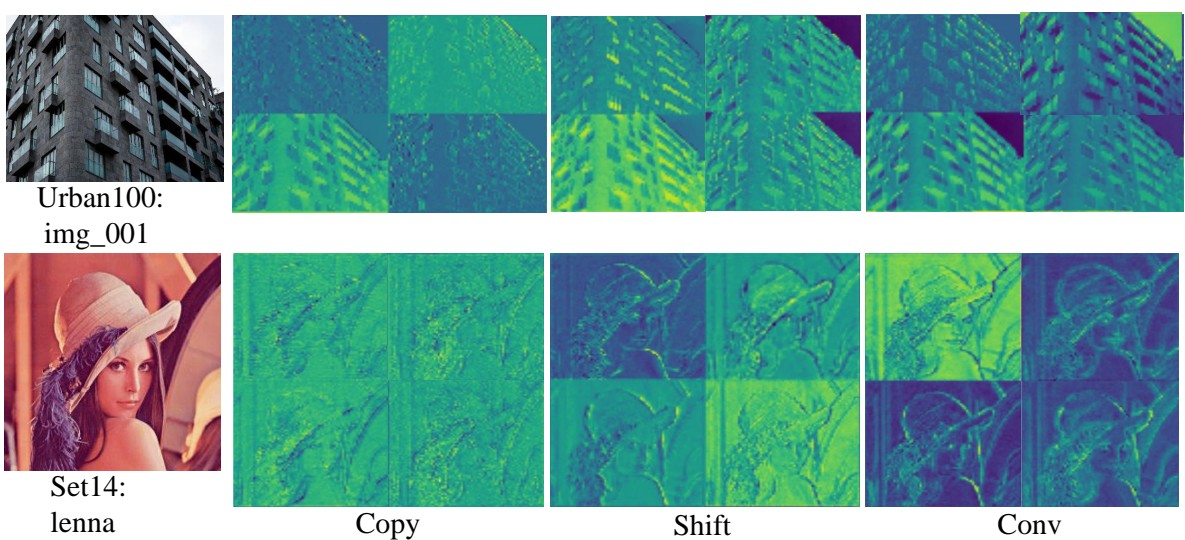

Figure 7: Visualization of features generated at the same layer by different CARN. The learnable shift operation extracts more textures than copy.

### 4.5.2 Analysis on the Ratio of Ghost Features.

We also conduct the experiments to analyze the effect of different ratio $\lambda$ of ghost features, and the results are reported in Table 6. A higher $\lambda$ means more ghost features are generated by the shift operation. Since the number of features in each layer is fixed, the more ghost features are generated by shift operation, the greater the reduction in parameters, FLOPs and latency, and the slight negative impact on PSNR/SSIM is inevitable. Notably, when the ratio equals 0.25, we achieve a higher PSNR on all four datasets with fewer parameters, FLOPs and latency.

Table 6: Analysis on the ratio of ghost features for ×3 CARN network.

| $\lambda$ | Params (M) | FLOPs (G) | Latency (ms) | Set5 PSNR/SSIM | Set14 PSNR/SSIM | B100 PSNR/SSIM | Urban100 PSNR/SSIM |
|---|---|---|---|---|---|---|---|
| 0.00 | 1.59 | 119 | 32.67 | 34.35/0.9266 | 30.32/0.8415 | 29.08/0.8043 | 28.05/0.8499 |
| 0.25 | 1.33 | 96 | 29.22 | ↑0.04/↑0.0003 | ↑0.05/↑0.0008 | ↑0.01/↑0.0000 | ↑0.04/↑0.0006 |
| 0.50 | 1.19 | 77 | 26.14 | ↓0.08/↓0.0004 | ↓0.02/↓0.0005 | ↓0.01/↓0.0004 | ↓0.08/↓0.0019 |
| 0.75 | 1.01 | 60 | 22.54 | ↓0.15/↓0.0021 | ↓0.08/↓0.0016 | ↓0.10/↓0.0018 | ↓0.23/↓0.0047 |

## 5 Conclusion

This paper proposes a GhostSR method for efficient SISR models. We first study the feature redundancy in convolutional layers and introduce a learnable shift operation to replace a part of conventional filters for generating the ghost features in a cheap way. Then, we present a procedure of clustering pre-trained models to select the intrinsic filters for generating intrinsic features. Thus, we can effectively learn and identify the intrinsic and ghost features simultaneously. The final output features are constructed by concatenating the intrinsic and ghost features together. We empirically analyze the benefits of shift operation on SISR task and bring a practical inference acceleration on GPUs. Extensive experiments demonstrate that both the non-compact and lightweight SISR models embedded in the GhostSR method can achieve comparable quantitative result and qualitative quality to that of their baselines with a large a large reduction of parameters, FLOPs and GPU inference latency.

## 6 Acknowledgement

We gratefully acknowledge the support of MindSpore (Huawei, 2020), CANN(Compute Architecture for Neural Networks) and Ascend AI Processor used for this research.

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

# A  Appendix

## A.1  Detailed Network Architectures

As we mentioned in the main body, to improve the performance of the models equipped with GhostSR, we do not replace the regular convolution in the first and the last layers in these networks, and the point-wise convolution is also kept unchanged if encountered in the middle layers. We take the ×2 scale as an example and the modified network architectures are summarized in Table 7, Table 8, Table 9 and Table 10. Where (cin, cout, k, k) in layer_size represents the input channel, output channel and kernel size of the convolutional layer are cin, cout and k × k, respectively. [c1_conv, c2_ghost] in type represents the output channel of conventional convolution operation and ghost operation in GhostSR method are c1 and c2, respectively. Note that when c2 is equal to 0, the convolutional layer only involves the conventional convolution operation.

Table 7: The detailed architecture of ×2 EDSR (Lim et al., 2017) equipped with GhostSR.

| layer_name | layer_size | type |
|---|---|---|
| head | (3, 256, 3, 3) | [256_conv, 0_ghost] |
| [ResBlock] × 32 | (256, 256, 3, 3) | [128_conv, 128_ghost] |
| tail.upsampler | (256, 1024, 3, 3) | [1024_conv, 0_ghost] |
| tail.conv | (256, 3, 3, 3) | [3_conv, 0_ghost] |

Table 8: The detailed architecture of ×2 RDN (Zhang et al., 2018c) equipped with GhostSR.

| layer_name | layer_size | type |
|---|---|---|
| SFENet1 | (3, 64, 3, 3) | [64_conv, 0_ghost] |
| SFENet2 | (64, 64, 3, 3) | [32_conv, 32_ghost] |
| [ResDenseBlock] × 16 | $(64 \times i, 64, 3, 3)$ $i = [1, 2, ... , 8]$ | [32_conv, 32_ghost] |
| GFF.conv0 | (1024, 64, 1, 1) | [64_conv, 0_ghost] |
| GFF.conv1 | (64, 64, 3, 3) | [64_conv, 0_ghost] |
| UPNet.conv0 | (64, 256, 3, 3) | [256_conv, 0_ghost] |
| UPNet.conv1 | (64, 3, 3, 3) | [3_conv, 0_ghost] |

Table 9: The detailed architecture of ×2 CARN (Ahn et al., 2018) equipped with GhostSR.

| layer_name | layer_size | type |
|---|---|---|
| conv_in | (3, 64, 3, 3) | [64_conv, 0_ghost] |
| Block1 | (64, 64, 3, 3) | [32_conv, 32_ghost] |
| C1 | (128, 64, 1, 1) | [64_conv, 0_ghost] |
| Block2 | (64, 64, 3, 3) | [32_conv, 32_ghost] |
| C2 | (192, 64, 1, 1) | [64_conv, 0_ghost] |
| Block3 | (64, 64, 3, 3) | [32_conv, 32_ghost] |
| C3 | (256, 64, 1, 1) | [64_conv, 0_ghost] |
| upconv | (64, 256, 3, 3) | [256_conv, 0_ghost] |
| conv_out | (64,3,3,3) | [3_conv, 0_ghost] |

Table 10: The detailed architecture of ×2 IMDN (Hui et al., 2019) equipped with GhostSR.

| layer_name | layer_size | type |
|---|---|---|
| conv1 | (3, 64, 3, 3) | [64_conv, 0_ghost] |
| [IMDModule] × 6 | C1 (64, 64, 3, 3) | [32_conv, 32_ghost] |
| | C2 (48, 64, 3, 3) | [32_conv, 32_ghost] |
| | C3 (48, 64, 3, 3) | [32_conv, 32_ghost] |
| | C4 (48, 16, 3, 3) | [8_conv, 8_ghost] |
| | C5 (64, 64, 1, 1) | [64_conv, 0_ghost] |
| C | (384, 64, 1, 1) | [64_conv, 0_ghost] |
| conv2 | (64, 64, 3, 3) | [64_conv, 0_ghost] |
| upsampler | (64, 12, 3, 3) | [12_conv, 0_ghost] |

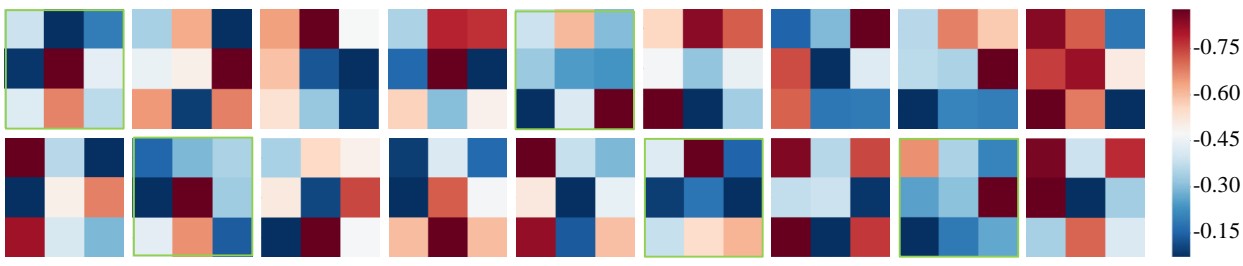

Figure 8: Visualization of the normalized depthwise filters.

## A.2 Detailed Results of the Baseline Convolutional Networks

For a fair comparison, we report the results of the baseline convolutional networks in Table 11, including the results of our re-trained and the results in the original papers. The results of our re-trained are close to the results in the original papers.

## A.3 Generalibity on Other Low-Level Tasks

In general, the proposed method is also applicable to other non-SISR low-level tasks. We conduct experiments on the SADNet (Chang et al., 2020) for the single image denoising task. Specifically, only the $3 \times 3$ conventional convolution filters are replaced with the Ghost version, and the convolution filters in the first and last layers are also kept unchanged. The results on the synthetic color BSD68 dataset with $\sigma = 30$ gaussian noise are reported in Table 12. The FLOPs are computed based on $480 \times 320$ color images. From the results in Table 12, the Ghost version significantly reduces parameters and FLOPs of original SADNet by 1.3M and 14.4G, respectively, while the PSNR is slightly reduced by 0.09 $dB$.

## A.4 Visualization of Depthwise Filters

In Table 4, we compare the results of replacing the shift operation in the proposed GhostSR method with the depthwise convolution. From Figure 1, the output features ($O$ in short) are obtained by concating the intrinsic features ($I$ in short) and ghost features ($G$ in short) together, that is, $O = [I, G]$. For the shift operation, $G = Shift(I)$, and for the depthwise convolution operation (DWConv in short), $G = DWConv(I)$. To compare the differences between shift and depthwise convolution, the learned depthwise convolution filters in the first GhostSR layer are randomly selected for visualization. Specifically, for a filter $W \in \mathbb{R}^{3 \times 3}$, every element $w$ of $W$ is first normalized by $|w|/max(|W|)$, and the normalized depthwise convolution filters are visualized in Figure 8. From Figure 8, we can see that some filters (marked with green rectangle) are very

Table 11: The detailed results of the baseline convolutional networks for scaling factor ×2, ×3 and ×4.

| Scale | Model | Type | Set5 PSNR/SSIM | Set14 PSNR/SSIM | B100 PSNR/SSIM | Urban100 PSNR/SSIM |
|---|---|---|---|---|---|---|
| ×2 | EDSR | Ours | 38.22 / 0.9612 | 33.86 / 0.9201 | 32.34 / 0.9018 | 32.92 / 0.9356 |
| | | Paper | 38.11 / 0.9601 | 33.92 / 0.9195 | 32.32 / 0.9013 | 32.93 / 0.9351 |
| | RDN | Ours | 38.25 / 0.9614 | 33.97 / 0.9205 | 32.34 / 0.9017 | 32.90 / 0.9355 |
| | | Paper | 38.24 / 0.9614 | 34.01 / 0.9212 | 32.34 / 0.9017 | 32.89 / 0.9353 |
| | CARN | Ours | 37.88 / 0.9601 | 33.54 / 0.9173 | 32.14 / 0.8990 | 31.96 / 0.9267 |
| | | Paper | 37.76 / 0.9590 | 33.52 / 0.9166 | 32.09 / 0.8978 | 31.92 / 0.9256 |
| | IMDN | Ours | 37.91 / 0.9595 | 33.59 / 0.9170 | 32.15 / 0.8988 | 32.14 / 0.9275 |
| | | Paper | 38.00 / 0.9605 | 33.63 / 0.9177 | 32.19 / 0.8996 | 32.17 / 0.9283 |
| ×3 | EDSR | Ours | 34.67 / 0.9293 | 30.56 / 0.8466 | 29.26 / 0.8094 | 28.78 / 0.8649 |
| | | Paper | 34.65 / 0.9282 | 30.52 / 0.8462 | 29.25 / 0.8093 | 28.80 / 0.8653 |
| | RDN | Ours | 34.69 / 0.9296 | 30.54 / 0.8467 | 29.26 / 0.8093 | 28.79 / 0.8654 |
| | | Paper | 34.71 / 0.9296 | 30.57 / 0.8468 | 29.26 / 0.8093 | 28.80 / 0.8653 |
| | CARN | Ours | 34.35 / 0.9266 | 30.32 / 0.8415 | 29.08 / 0.8043 | 28.05 / 0.8499 |
| | | Paper | 34.29 / 0.9255 | 30.29 / 0.8407 | 29.06 / 0.8034 | 28.06 / 0.8493 |
| | IMDN | Ours | 34.32 / 0.9260 | 30.31 / 0.8410 | 29.07 / 0.8037 | 28.15 / 0.8511 |
| | | Paper | 34.36 / 0.9270 | 30.32 / 0.8417 | 29.09 / 0.8046 | 28.17 / 0.8519 |
| ×4 | EDSR | Ours | 32.46 / 0.8984 | 28.81 / 0.7874 | 27.73 / 0.7416 | 26.60 / 0.8019 |
| | | Paper | 32.46 / 0.8968 | 28.80 / 0.7876 | 27.71 / 0.7420 | 26.64 / 0.8033 |
| | RDN | Ours | 32.49 / 0.8987 | 28.83 / 0.7875 | 27.72 / 0.7416 | 26.59 / 0.8023 |
| | | Paper | 32.47 / 0.8990 | 28.81 / 0.7871 | 27.72 / 0.7419 | 26.61 / 0.8028 |
| | CARN | Ours | 32.16 / 0.8943 | 28.59 / 0.7810 | 27.58 / 0.7355 | 26.03 / 0.7830 |
| | | Paper | 32.13 / 0.8937 | 28.60 / 0.7806 | 27.58 / 0.7349 | 26.07 / 0.7837 |
| | IMDN | Ours | 32.19 / 0.8938 | 28.57 / 0.7805 | 27.54 / 0.7344 | 26.03 / 0.7831 |
| | | Paper | 32.21 / 0.8948 | 28.58 / 0.7811 | 27.56 / 0.7353 | 26.04 / 0.7838 |

Table 12: Comparison of the baseline SADNet (Chang et al., 2020) and the Ghost version.

| Model | Params (M) | FLOPs (G) | PSNR ($dB$) |
|---|---|---|---|
| Original | 4.3 | 50.1 | 30.64 |
| Ghost | 3.0 | 35.7 | 30.55 |

similar to shift operation, that is, only one value of 3x3 is equal to 1, and the others are significantly low, even close to 0.

## A.5 Comparison with SwinIR

SwinIR Liang et al. (2021) is the representative work based on transformers, which achieves impressive results on many low-level vision tasks. The detailed comparisons are reported in Table 13. The average inference latency is calculated on Urban100 dataset with ×4 scale on a single V100 GPU platform. From Table 13, compared with other models, SwinIR achieves the highest PSNRs on four datasets, however, SwinIR also has the highest latency at an astounding 153.4ms. The ultra-high latency limits the deployment of SwinIR in real-time inference scenarios. The PSNRs of the 4x wider Ghost_IMDN model are comparable to SwinIR, however, the latency is only one-third of SwinIR.

Table 13: Comparison with SwinIR Liang et al. (2021) with×4 scale.

| Model | Params (M) | FLOPs (G) | Latency (ms) | Set5 PSNR | Set14 PSNR | B100 PSNR | Urban100 PSNR |
|---|---|---|---|---|---|---|---|
| SwinIR | 0.90 | 49.6 | **153.4** | 32.44 | 28.77 | 27.69 | 26.47 |
| IMDN | 0.72 | 40.9 | 8.6 | 32.19 | 28.57 | 27.54 | 26.03 |
| Ghost_IMDN | 0.42 | 23.9 | 6.9 | 32.06 | 28.49 | 27.48 | 25.89 |
| Ghost_IMDN (x4) | 6.47 | 370.5 | 50.12 | 32.40 | 28.71 | 27.61 | 26.38 |

### A.6  More Visual Results

In Figure 9, more results are displayed to reveal the difference of visual quality between the original network and the GhostSR versions. For both the non-compact and lightweight networks, the details and textures generated by GhostSR are basically the same as those by original network.

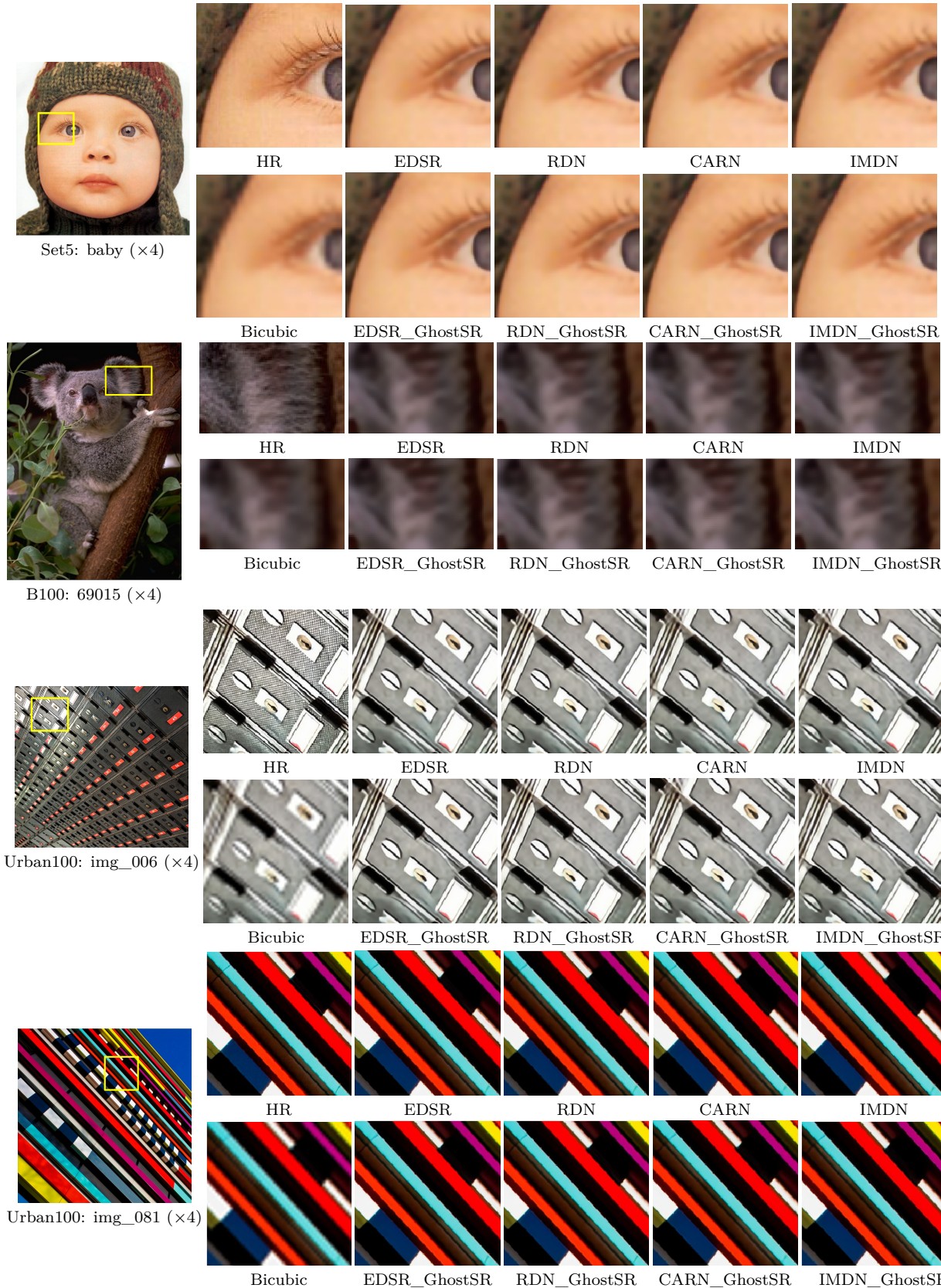

Figure 9: Visual comparisons for ×4 images on various datasets. For the shown examples, the details generated by ghost operation are approximately the same as those by conventional convolution.

