# OpenReview forum: "GhostSR: Learning Ghost Features for Efficient Image Super-Resolution"
_TMLR — Accepted by TMLR_

### Review · Reviewer_kXZf · 2022-08-30

**Summary Of Contributions:**

This paper investigates a problem on the feature redundancy of convolutional neural networks in the single image super-resolution (SISR) task. To address this, the authors propose an efficient SISR model by using shift operation to replace a part of conventional filters and generate the redundant features (i.e., ghost feature). To make the shift operation learnable, the authors use a Gumbel-Softmax trick and present a procedure of clustering pre-trained models to select the intrinsic filters for generating intrinsic features. Extensive experiments on some datasets demonstrate that both the noncompact
and lightweight SISR models embedded in the GhostSR method are able to achieve comparable performance with fewer parameters, FLOPs and GPU inference latency.

**Requested Changes:**

1. The motivation for using shift operation should be discussed in the introduction. Additional features are important in SISR. However, why only use shift operation in the model? The shift operation may reduce top left information, as shown in Figure 1. Moreover, convolution has a shift-invariant property and obtains the same amount of information. There are some augmentation operations (e.g., scaling and ColorJitter) in existing computer vision tasks. Besides, some augmentation operations can increase additional information, e.g., different scale information. It would be better to exploit other augmentation operations to improve SR performance.

2. In Figure 2, the benefits of shifting the features for the SISR task are not clear. (a) Small shifts can obtain high-frequency gradient information.  Is such gradient information concatenated in the features? (b) The shift operation seems cannot enlarge the receptive field of filters when the kernel size is fixed.

3. Some baseline methods should be compared. For example, the authors should compare a baseline model by reducing the convolutional layer to a comparable parameter size and train this model with raw features.

4. Different datasets have different performance gains. In Table 1, the model trained on Urban100 has a larger performance drop than other test sets. Could you please give more discussions in the paper?

5. The compared state-of-the-arts are old. The authors should compare with recent SOTA methods (e.g., SwinIR) and conduct experiments by learning Ghost features.

6. In Table 3, the performance of GhostSR is not significant. Compared with GhostSR, the Lightweight SwinIR model has a smaller model size of 886K and a higher PSNR of 28.66 db on Urban100.

7. In Figure 5, it would be better to plot PSNR v.s. FLOPs v.s. Model size. In Figure 7, why shift operation can generate distinguishing feature maps? More discussions should be provided.

**Strengths And Weaknesses:**

Strengths:
1. This paper investigates the feature redundancy of convolutional neural networks in the SISR task.
2. This paper analyzes the benefits of shift operation on the SISR task and accelerates the practical inference on GPUs.
3. Extensive experiments demonstrate that learning Ghost features helps achieve comparable performance with fewer parameters, FLOPs and GPU inference latency.

Weaknesses:
1. Some motivations should be highlighted in this paper.
2. The model uses shift operation and Gumbel-Softmax trick is not novel.
3. Some technical details of the proposed method are not clear.
4. The experiment section should be improved in the paper.

More details can be found in the next section.

---

### Review · Reviewer_yxaN · 2022-09-06

**Summary Of Contributions:**

The paper leverages shift-based operation for generating ghost features for SISR tasks to yield benefits on hardware like GPUs. The paper then introduced learnable shift operations via Gumble softmax operator. The authors provided results on four standard benchmark datasets: set5, set14, B100, and Urban100 to show the efficacy of their proposed approach.

**Broader Impact Concerns:**

No.

**Requested Changes:**

The authors need to add significant amount of contributions to motivate a new work, the current work of efficient convolution for ghost feature extraction is largely taken from existing literature.

**Strengths And Weaknesses:**

#### Strengths
============

1. The paper is easy to follow and well motivated. The demand for reduced complexity SISR model is interesting as the authors mentioned that they require even more FLOPs compared to standard classification tasks.

2. The application of shift operations instead of DWC is well motivated as such DWCs often suffer from latency disadvantages in hardware like GPUs.

3. There are some good results with the learnable shift operations.

#### Weaknesses
============

1. The main weakness of the paper is its contribution, the authors have leveraged already published work to solve similar problem of DNNs, that is the complexity reduction of convolution operation.

2. With the exhaustive research in the domain of Shift-add based CONV operations, for example: shift-add Net [1], Shift-add NAS [2], already showed efficacy of such approaches. This reduces the contribution and insights of the current manuscript.

3. The implementation of Gumble softmax to make the opeation learnable is not new either as already mentioned by the authors.

Thus overall, I think its a good work with not enough new contribution and thus validating some of the well accepted efficient convolution approximations.

[1] ShiftAddNet: A Hardware-Inspired Deep Network, NeurIPS 2020.
[2] ShiftAddNAS: Hardware-Inspired Search for More Accurate and Efficient Neural Networks, ICML 2022.

---

### Review · Reviewer_XCf9 · 2022-09-10

**Summary Of Contributions:**

This paper presents a method to make single image super-resolution models more efficient, by replacing a subset of convolution filters with discrete shift operations.  Observing that intermediate layer feature maps can be very similar, after accounting for shifts this redundancy can be leveraged by reducing the number of convolution operations in favor of shifted replication.  The (x,y) shift amounts for each filter are learned using Gumbel-softmax on a dxd matrix of possible shifts.  To decide how many shifts to allow for each filter, two options are explored: (1) train from scratch with one shift for each filter (in lambda=0.5 case), or (2) cluster the learned filters in a pretrained SISR model, and assign as many shifts to each intrinsic filter as there elements in that filter's cluster.  Both methods arrive at good performance, with preinitializing better.  The method is evaluated on four datasets, incorporating into numerous existing SISR methods, finding about 40% reduction in flops for near-indistinguishable performance differences.


**Requested Changes:**

Please see above questions.  In my view, these could further strengthen the paper; none are critical.

**Strengths And Weaknesses:**

This is a sound approach applied in the context of SISR, identifying a fast and simple operation that can replace more expensive multiply-add operations that learn similar features anyway due to low-level nature of the SISR task.  For the most part it is clearly described.  The performance of the method is robustly demonstrated with evaluations on multiple SISR methods and datasets.

I have a few questions, which I think could strengthen the paper further:


- Although the paper discusses feature redundancy, I don't see any measurements or explicit investigation of this.  Better establishing there is indeed shift redundancy before using this method would more precisely confirm the motivation.  A few possible measures:  How close are filters or feature maps to each other, taking minimum over all kxk shift offsets?  Or, if a SISR network uses depthwise-channelwise decomposition of a conv layer, how many of the depthwise conv filters are close to shifted copies of another filter?

- What is offset range d used for W  (I did not see it in the text or appendix, but may have missed it)

- How are offset weights W initialized?  For pretrained model filter clustering init: Are the shift weights W also initialized finding the shifts that best approximate the other (unselected) filters in each cluster from the selected one, using an initial greedy search over all shifts?  Or are they random?

- 4.4.2:  this is an interesting evaluation, I would also be interested in seeing what the depthwise conv filters look like --- are a lot of them close to point shifts?


- 4.5.1:  "we replace the shift operation by simple copy":  does this mean the features are copied directly with no shift at all (so they are the same features at the same offsets), or that there are shifts but these are not learned?  If there is no shift, how does this compare to removing the features entirely?  What is the next downstream op, and can it benefit from unshifted replicated features (if it's a linear conv layer, I don't see how it would benefit from copies).


- While clear, the English writing is rough in several places, particularly in the abstract, which does not read very fluidly.  There are some unusual word choices and number/tense agreements that are off here, too.  This is not a major concern, as both the main body and abstract are still clear.  However, I would recommend trying to edit the abstract over again keeping this in mind.

---

### Review · Reviewer_mJsT · 2022-09-12

**Summary Of Contributions:**

Summary:

This paper presents a new method for efficient single image super-resolution with the help of redundant features referred to as ghost features generated by the shift operation. To make the ghost features informative, the authors proposed the learnable shift operation based on Gumbel-softmax for diffusional computation. The original pre-trained model has been compressed by the clustering of filters, with the centroid filters preserved. The final model includes both intrinsic and ghost features for joint representation learning.

**Broader Impact Concerns:**

No.

**Requested Changes:**

Cons:

- The major concern is the novelty since GhostNet has already been proposed for high-level vision tasks, which makes this work incremental (like a simple extension for low-high vision tasks). The discussion of the difference between GhostNet and this work is not clear enough. On page 2, I am still confused why the “so-called cheap operation is not cheap”? Could the authors explain more about this point?
- The minor concern is the visual effectiveness since I have not seen the significant improvements in Fig 4. More visual examples are welcome to be included.
- Some important refs in the efficient SISR area are missing [1-3].

[1] Learning Efficient Image Super-Resolution Networks via Structure-Regularized Pruning. ICLR 21.

[2] Aligned Structured Sparsity Learning for Efficient Image Super-Resolution. NeurIPS 21.

[3] NTIRE 2022 Challenge on Efficient Super-Resolution: Methods and Results. CVPRW 22.

- K-means clustering for filters is not very convincing for me. The authors are suggested to discuss more of why choosing K-means and why it works.

Others:

In Eq. 4, why W^{'}{i,j} + N{i,j}?

Duplicate references for the paper “Towards lightweight image super-resolution with lattice block”.

**Strengths And Weaknesses:**

Pros:

+ The idea of creating redundant features as augmentation of the original model is pretty interesting. Especially in SISR, the high-frequency features are crucial and can be enhanced by ghost features. Therefore, the motivation is strong.

+ The experimental results are competitive in efficiency for both model size compression and inference speed acceleration.

+ This proposed method is agnostic to different pre-trained models based on CNN.

+ The whole paper is well written, with clear logic to follow.

---

### Decision · Action_Editors · 2022-10-22

**Recommendation:** Accept with minor revision

**Comment:**

This manuscript develops a relatively efficient image super-resolution method, in the family of deep learning based ones, which uses a CNN equipped with learnable shift operations to generate the ghost features involved in the SR process. The proposed method has been demonstrated to achieve practical inference acceleration on GPU devices. The AE and three reviewers recognize the main technical contribution that leads to a non-trivial gain in the computational efficiency of SR.

To be accepted by TMLR, the AE would ask the authors to strengthen their work to address the following issues:
1)  Proofread the entire paper to fix typos and grammatical errors.
2)  Give more explanations on Figure 8 which showcases depthwise convolution kernels.
3)  Add a fair comparison against Lightweight SwinIR, which is vital and requested by the reviewer.
4)  Make your technical contribution more insightful other than just stating the hardware benefits.


**Audience:**

Yes.

**Claims And Evidence:**

Yes.